# ASK1 inhibits browning of white adipose tissue in obesity

Fabrizio C. Lucchini[1,2,3], Stephan Wueest [1,2], Tenagne D. Challa[1,2], Flurin Item[1,2], Salvatore Modica[4], Marcela Borsigova[1,2], Yulia Haim[5,6], Christian Wolfrum[4], Assaf Rudich[5,6] & Daniel Konrad [1,2,3✉]

Increasing energy expenditure via induction of adipose tissue browning has become an appealing strategy to treat obesity and associated metabolic complications. Herein, we identify adipocyte-expressed apoptosis signal-regulating kinase 1 (ASK1) as regulator of adipose tissue browning. High fat diet-fed adipocyte-specific ASK1 knockout mice reveal increased UCP1 protein levels in inguinal adipose tissue concomitant with elevated energy expenditure, reduced obesity and ameliorated glucose tolerance compared to control litter-mates. In addition, ASK1-depletion blunts LPS-mediated downregulation of isoproterenol-induced UCP1 in subcutaneous fat both in vitro and in vivo. Conversely, adipocyte-specific ASK1 overexpression in chow-fed mice attenuates cold-induced UCP1 protein levels in inguinal fat. Mechanistically, ASK1 phosphorylates interferon regulatory factor 3 (IRF3) resulting in reduced *Ucp1* expression. Taken together, our studies unravel a role of ASK1 in mediating the inhibitory effect of caloric surplus or LPS-treatment on adipose tissue browning. Adipocyte ASK1 might be a pharmacological target to combat obesity and associated morbidities.

[1] Division of Pediatric Endocrinology and Diabetology, University Children's Hospital, CH-8032 Zurich, Switzerland. [2] Children's Research Center, University Children's Hospital, CH-8032 Zurich, Switzerland. [3] Zurich Center for Integrative Human Physiology, University of Zurich, CH-8057 Zurich, Switzerland. [4] Institute of Food, Nutrition and Health, ETH Zurich, CH-8603 Schwerzenbach, Switzerland. [5] Department of Clinical Biochemistry and Pharmacology, Ben-Gurion University of the Negev, 84103 Beer-Sheva, Israel. [6] The National Institute of Biotechnology in the Negev, Ben-Gurion University of the Negev, 84103 Beer-Sheva, Israel. ✉email: daniel.konrad@kispi.uzh.ch

Dissipation of energy in brown adipose tissue (BAT) has emerged as a promising target to combat obesity and associated co-morbidities[1–4]. It depends on uncoupling protein 1 (UCP1)-mediated proton leak across the inner mitochondrial membrane leading to heat production[5]. Besides BAT, beige adipocytes residing in white adipose tissue (WAT) can express UCP1, and its expression is induced by β3-adrenergic receptor stimulation leading to browning of WAT[6]. Given the fact that adult humans have little or no existing active BAT, selective induction of WAT browning rather than activation of BAT is thought to have therapeutic potential to counteract the obesity pandemic[7,8]. However, while many factors affecting browning of WAT have been described in rodents, only very few revealed a selective effect constrained to WAT[9].

Obesity is associated with a chronic low-grade inflammation of adipose tissue characterized by elevated local and circulating levels of pro-inflammatory cytokines and endotoxins[10,11]. Interestingly, recent reports suggest that these inflammatory factors may negatively affect browning of WAT. In fact, the pro-inflammatory cytokine tumor necrosis factor-alpha (TNFα) and the endotoxin lipopolysaccharide (LPS), which are both elevated in circulation as well as in adipose tissue under obesity[12–15], reduced UCP1 levels in adipocytes[16–18].

Strikingly, TNFα as well as LPS are activators of the apoptosis signal-regulating kinase 1 (ASK1)[4,19,20]. In fact, ASK1 is specifically activated by LPS and selectively required for TLR4 signaling[19]. ASK1 is a family member of the mitogen-activated protein kinase kinase kinase (MAP3K)[21], acting as a signaling node in which different stressors such as endoplasmic reticulum, oxidative and inflammatory stresses converge[4]. In addition to being activated by different stressors, ASK1 signaling can be induced via autophosphorylation[22] suggesting that a mere increase in ASK1 expression results in elevated ASK1 activity. Importantly, elevated ASK1 expression in adipose tissue of obese human subjects was found to be an independent predictor of whole-body insulin resistance[23], underscoring the potential importance of this stress-signaling pathway in obesity and associated metabolic disorders. Herein, we hypothesize that obesity-induced expression/activation of ASK1 in adipocytes negatively affects UCP1 protein expression in WAT thereby mediating an inhibitory effect on adipose tissue browning.

## Results

### LPS treatment blunts Ucp1 expression in subcutaneous adipocytes ASK1-dependently.

Activation of the TLR4 pathway via chronic injection of LPS decreased cold-induced expression of UCP1 in subcutaneous WAT of C57BL/6 mice[16]. In line, we found that chronic LPS administration using osmotic mini-pumps significantly reduced UCP1 protein levels in inguinal WAT of C57BL/6 mice that were cold-exposed for 7 days (Fig. 1a, b), a condition previously shown to induce browning of WAT in vivo[24]. In addition, LPS treatment blunted β3-adrenoreceptor agonist-induced Ucp1 mRNA expression in inguinal fat explants harvested from lean C57BL/6 mice (Fig. 1c), further supporting a negative impact of LPS on the thermogenic potential of WAT. To elucidate the molecular mechanism, experiments in immortalized pre-adipocytes derived from murine subcutaneous WAT[25] were performed. As expected, treatment of mature subcutaneous (sc) white adipocytes with the β3-adrenoreceptor agonist isoproterenol increased Ucp1 mRNA expression levels (>100 fold) compared to untreated control cells (Supplementary Fig. 1a) indicating a high browning capacity of these cells. Similar to the effect in inguinal fat explants (Fig. 1c), pre-treatment of subcutaneous white adipocytes with LPS significantly reduced isoproterenol-induced Ucp1 expression by ~50 % (Fig. 1d)

suggesting that such cells may be a valid model to investigate molecular mechanisms involved in LPS-induced inhibition of browning.

As outlined above, ASK1 is an important signaling node downstream of LPS as well as other mediators and might be involved in the suppression of obesity-induced WAT browning[17–19,26]. Of note, ASK1 levels in adipose tissue were increased in obesity[23] suggesting that elevated ASK1 expression may negatively affect UCP1 levels in adipose tissue. In line with this human data, we found significantly elevated ASK1 protein levels in inguinal adipose tissue of C57BL/6 mice fed a high-fat diet (HFD) compared to chow-fed control mice (Fig. 1e, f). In parallel, phosphorylated levels of ASK1 (Thr845) were significantly elevated in HFD-fed mice, indicating increased activation of ASK1 in the latter (Supplementary Fig. 1b). Moreover, treatment of subcutaneous white adipocytes with LPS increased Ask1 expression (Fig. 1g) similar to effects induced by TNFα, Fas ligand or oxidative stress[26]. Besides, LPS significantly induced phosphorylation of ASK1 in subcutaneous adipocytes as well as in inguinal WAT of C57BL/6 mice receiving chronic LPS administration using osmotic mini-pumps (Supplementary Fig. 1c, d). To test whether ASK1 may mediate LPS-induced blunting of Ucp1 expression (Fig. 1d), we generated a lentiviral construct expressing a short hairpin RNA (shRNA) against ASK1 (shASK1) to knockdown ASK1 in subcutaneous adipocytes. As intended, expression of Ask1 was significantly reduced in shASK1 transfected cells when compared to shLuc transfected control cells (Supplementary Fig. 1e). Strikingly, ASK1 knockdown significantly blunted LPS-induced Ucp1 downregulation (Fig. 1h), whereas shLuc transfected subcutaneous white adipocytes showed a similar degree of LPS-mediated reduction of Ucp1 mRNA expression compared to non-transfected cells (Fig. 1d). These results indicate that ASK1 is involved in the control of Ucp1 expression in subcutaneous adipocytes and, hence, may be a modulator of adipose tissue browning and, consequently, obesity.

### Reduced body weight and improved glucose tolerance in HFD-fed ASK1$^{\Delta adipo}$ mice.

To investigate whether depletion of adipocyte-expressed ASK1 affects browning and obesity, we generated adipocyte-specific ASK1 knockout mice (ASK1$^{\Delta adipo}$) on a C57BL/6 background using the Cre-lox system. As shown in the gene targeting design (Supplementary Fig. 2a), exon 14 of ASK1 (whose deletion leads to a frameshift) is removed in these mice upon Cre-recombinase (Cre) expression. As controls, littermate mice with floxed ASK1 but lacking Cre expression under the adiponectin promotor were used (ASK1$^{F/F}$). Adipocyte-specific deletion of ASK1 was confirmed by western blot analysis (Fig. 2a). As expected, we detected reduced ASK1 protein levels in BAT due to adiponectin expression in this tissue, while similar ASK1 protein levels were found in muscle, brain and liver (Supplementary Fig. 2b). In order to investigate the physiological significance of adipocyte-specific ASK1 depletion, mice were fed either a standard chow or HFD for 12 weeks. While body weight and weight gain was not affected in chow-fed mice (Fig. 2b, c), ASK1$^{\Delta adipo}$ mice fed a HFD revealed significantly reduced weight gain resulting in lower body weight compared to control littermates (Fig. 2b, c). Next, glucose tolerance tests were performed to assess whether adipocyte-specific depletion of ASK1 affected glucose metabolism. As expected, 12 weeks of HFD significantly deteriorated glucose tolerance in ASK1$^{F/F}$ mice compared to chow-fed mice (Fig. 2d, e). Importantly, glucose tolerance was significantly improved in HFD-fed ASK1$^{\Delta adipo}$ compared to control littermates (Fig. 2d, e). We used a short-term, 4 days HFD paradigm to differentiate between the effect of ASK1 deletion in adipocytes on weight from its effect on glucose tolerance. Even

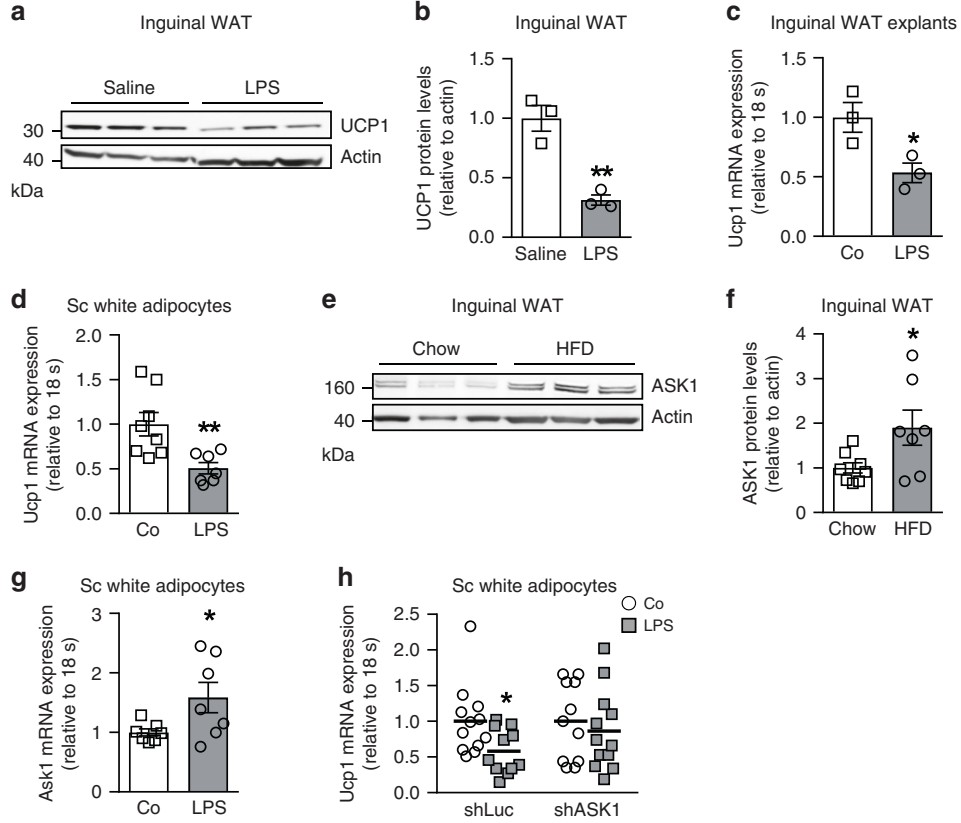

**Fig. 1 LPS treatment blunts UCP1 expression in subcutaneous adipocytes ASK1-dependently. a**, **b** Western blot and quantified protein levels of UCP1 in total lysates of inguinal white adipose tissue harvested from cold-exposed C57BL/6J mice chronically treated with or without LPS. $n = 3$ mice per group. **$p = 0.004$. **c** *Ucp1* mRNA expression in inguinal fat explants harvested from C57BL/6J mice were treated with or without 100 ng/ml LPS and subsequently with 10 μM isoproterenol for 8 h. $n = 3$ biological replicates. *$p = 0.036$. **d** *Ucp1* mRNA expression in subcutaneous adipocytes pre-treated with or without 100 ng/ml LPS for 24 h followed by stimulation with 0.1 μM isoproterenol for 6 h. $n = 7$ (LPS) or $n = 8$ (control) biological replicates. **$p = 0.007$. **e**, **f** Western blot and quantified protein levels of ASK1 in total lysates of inguinal white adipose tissue harvested from C57BL/6J mice fed a chow or HFD for 20 weeks. $n = 8$ (chow) and $n = 7$ (HFD) mice per group. *$p = 0.037$. **g** *Ask1* mRNA expression in subcutaneous adipocytes treated with or without 100 ng/ml LPS treatment for 24 h. $n = 7$ biological replicates. *$p = 0.046$. **h** *Ucp1* mRNA expression in subcutaneous adipocytes transfected with control shRNA lentivirus (shLuc) or shRNA lentivirus targeting ASK1 (shASK1) pre-treated with 100 ng/ml LPS for 24 h followed by stimulation with 0.1 μM isoproterenol for 6 h. $n = 11$ (shASK1 control and shLuc LPS) or n = 12 (shASK1 LPS and shLuc control) biological replicates. *$p = 0.027$. Values are expressed as mean ± SEM. Statistical tests used: two-sided *t*-tests for (**b–d**, **f–h**). Source data are provided as a Source Data file.

with no body weight difference after 4 days of HFD between the two groups (ASK1[F/F]: 21.6 ± 0.7 g vs. ASK1[Δadipo]: 21.2 ± 0.5 g; $p = 0.67$), glucose tolerance was improved in ASK1[Δadipo] (Supplementary Fig. 2c). Hence, depletion of ASK1 positively affects glucose metabolism independent of body mass under HFD conditions. In addition, circulating insulin levels were significantly reduced in HFD-fed knockout mice (Supplementary Table 1) indicating improved insulin sensitivity. To investigate the latter, hyperinsulinemic-euglycemic clamp studies were performed. A significantly increased glucose infusion rate in HFD-fed ASK1[Δadipo] compared to ASK1[F/F] mice was noted, consistent with improved whole-body insulin sensitivity (Fig. 2f and Supplementary Fig. 2d, e). Glucose uptake during clamp was significantly increased into inguinal and trend wise higher into epididymal adipose tissue of HFD-fed ASK1[Δadipo] compared to ASK1[F/F] mice. By contrast, glucose uptake into BAT and skeletal muscle was similar between both genotypes (Fig. 2g). Of note, endogenous glucose production under hyperinsulinemic clamp conditions was significantly reduced in ASK1[Δadipo] compared to control littermates indicating blunted hepatic insulin resistance (Fig. 2h). The latter is closely associated with visceral adipose tissue inflammation and liver steatosis[27]. As shown in Fig. 2i, HFD-fed ASK1[Δadipo] mice revealed significantly reduced hepatic

steatosis as determined by liver triglyceride accumulation. However, the degree of mesenteric fat inflammation, as assessed by mRNA expression of inflammatory markers such as *Tnfα*, *Il-6*, *Mcp1,* and *Il-1β*, did not significantly differ between the two genotypes (Fig. 2j). Moreover, expression of the anti-inflammatory cytokine *Il-10* and the macrophage marker *F4/80* were unchanged (Fig. 2j). In line, we found a similar degree of inflammation in inguinal adipose tissue of HFD-fed knockout and control mice (Supplementary Fig. 2f). In accordance, circulating levels of TNFα, IL-6, IL-10, IFN-γ and KC were comparable (Supplementary Table 1) suggesting a similar degree of adipose tissue inflammation in HFD-fed ASK1[Δadipo] and ASK1[F/F] mice.

Both adipocytes as well as the stromal vascular fraction, albeit to a lower degree, contributed to increased ASK1 expression in adipose tissue of obese human subjects[23]. Consequently, ASK1 activation in adipocytes as well as myeloid cells may play a role in obesity-induced metabolic complications. To investigate the contribution of myeloid cell-expressed ASK1 to the development of obesity and glucose intolerance, we generated myeloid cell-specific ASK1 knockout mice (ASK1flox/flox, Lysozyme-cre+/−; ASK1[Δmye]) using the Cre-lox system. Such Cre-mediated recombination results in depletion of ASK1 protein in myeloid

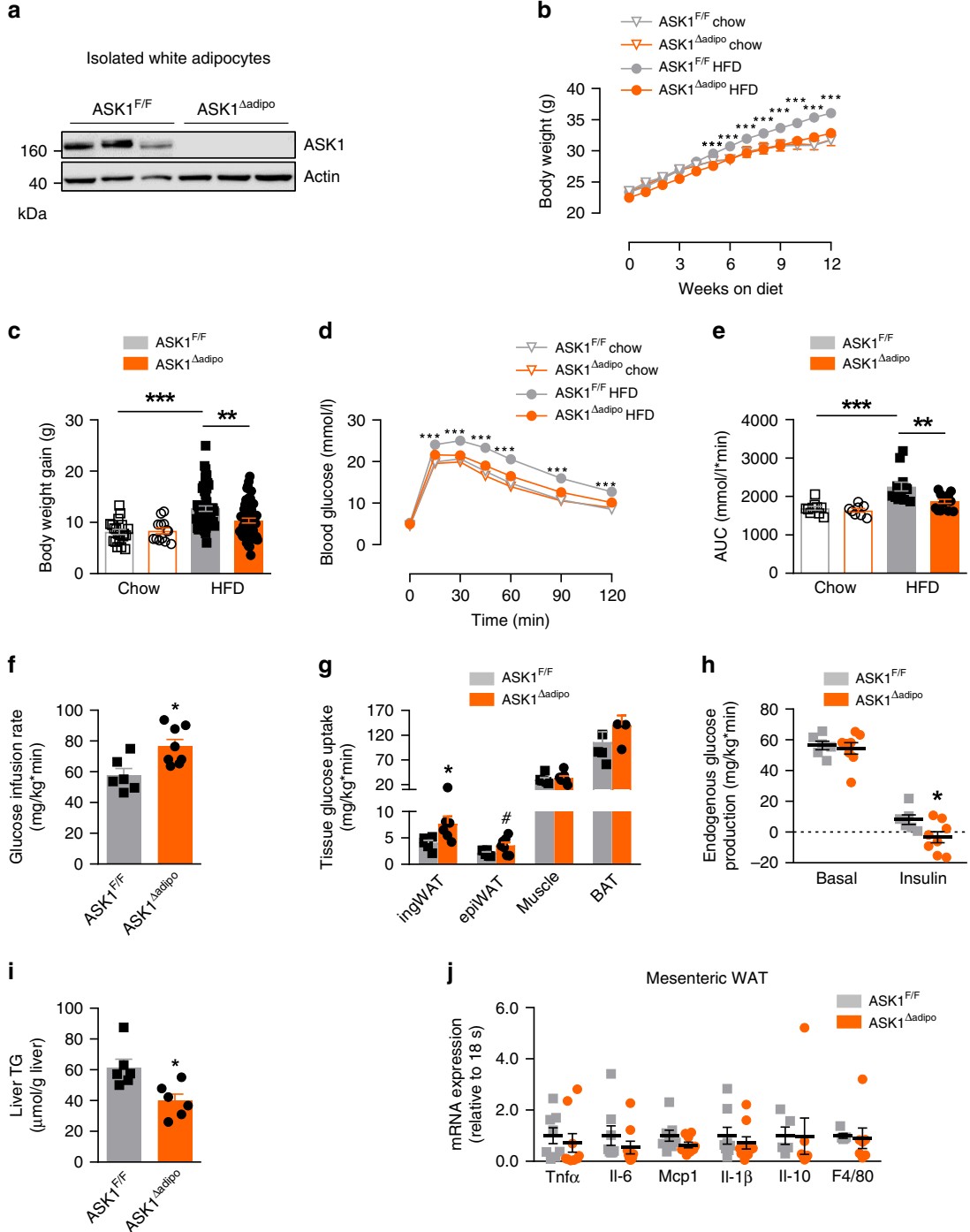

**Fig. 2 Reduced body weight and improved glucose tolerance in HFD-fed ASK1$^{\Delta adipo}$ mice. a** Protein levels of ASK1 in lysate of isolated epididymal adipocytes harvested from ASK1$^{F/F}$ and ASK1$^{\Delta adipo}$ mice ($n = 3$ mice per group). **b, c** Body weight and body weight gain during 12 weeks of chow (ASK1$^{F/F}$, $n = 20$ mice; ASK1$^{\Delta adipo}$, $n = 13$ mice) or HFD feeding (ASK1$^{F/F}$, $n = 52$ mice; ASK1$^{\Delta adipo}$, $n = 58$ mice). **$p = 0.02$, ***$p < 0.001$. **d, e** Intraperitoneal glucose-tolerance test (chow-fed: ASK1$^{F/F}$, $n = 10$ mice; ASK1$^{\Delta adipo}$, $n = 8$ mice; HFD-fed: ASK1$^{F/F}$, $n = 13$ mice; ASK1$^{\Delta adipo}$, $n = 14$ mice) at 18 weeks of age. **$p = 0.03$, ***$p < 0.001$. **f** Glucose infusion rate (GIR) during hyperinsulinemic-euglycemic clamps in HFD-fed ASK1$^{F/F}$ ($n = 6$) and ASK1$^{\Delta adipo}$ ($n = 8$) mice. *$p = 0.011$. **g** Glucose uptake into respective tissues during hyperinsulinemic-euglycemic clamps in HFD-fed ASK1$^{F/F}$ ($n = 5$ (ing WAT, epiWAT and skeletal muscle) or $n = 6$ (BAT)) and ASK1$^{\Delta adipo}$ ($n = 4$ (BAT), $n = 6$ (ingWAT) or $n = 7$ (epiWAT and skeletal muscle)) mice. *$p = 0.034$, #$p = 0.062$. **h** Endogenous glucose production (EGP) during hyperinsulinemic-euglycemic clamps in HFD-fed ASK1$^{F/F}$ ($n = 6$) and ASK1$^{\Delta adipo}$ ($n = 8$) mice. *$p = 0.040$. **i** Liver triglyceride levels determined in HFD-fed ASK1$^{F/F}$ ($n = 6$) and ASK1$^{\Delta adipo}$ ($n = 6$) mice. *$p = 0.012$. **j** Mesenteric adipose tissue mRNA expression of respective targets determined in HFD-fed ASK1$^{F/F}$ ($n = 5$ (Il-10, F4/80) or $n = 8$ (TNFα, Il-6, Mcp1, Il-1β) and ASK1$^{\Delta adipo}$ ($n = 7$ (Il-10, F4/80) or $n = 9$ (TNFα, Il-6, Mcp1, Il-1β)) mice. Values are expressed as mean ± SEM. Statistical tests used: two-sided t-tests for (**f, g**) (without adjustments for multiple comparisons), **h, i** ANOVA for (**b–e**). AUC: area under the curve. Source data are provided as a Source Data file.

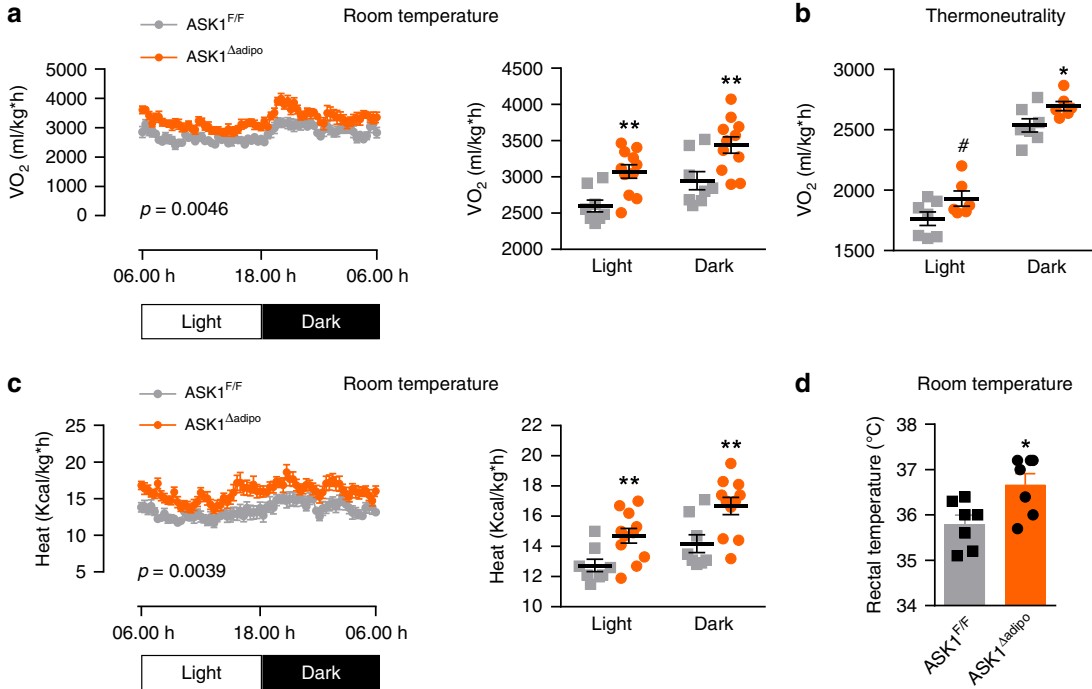

**Fig. 3 Increased heat production in HFD-fed ASK1$^{\Delta adipo}$ mice. a** Light (day) and dark (night) oxygen consumption in HFD-fed ASK1$^{F/F}$ ($n = 8$) and ASK1$^{\Delta adipo}$ ($n = 11$) mice during a 24-h period at RT. **$p = 0.020$ (light), **$p = 0.010$ (dark). **b** Light (day) and dark (night) oxygen consumption in HFD-fed ASK1$^{F/F}$ ($n = 7$) and ASK1$^{\Delta adipo}$ ($n = 6$) mice during a 24-h period after 20 days housing at thermoneutrality. #$p = 0.073$, *$p = 0.042$. **c** Light (day) and dark (night) heat production in HFD-fed ASK1$^{F/F}$ ($n = 8$) and ASK1$^{\Delta adipo}$ ($n = 11$) mice during a 24-h period at RT. **$p = 0.009$. **d** Rectal temperature measured in HFD-fed ASK1$^{F/F}$ ($n = 7$) and ASK1$^{\Delta adipo}$ ($n = 7$) mice. *$p = 0.015$. Values are expressed as mean ± SEM. Statistical tests used: two-sided $t$-tests for (**a**–**d**). Source data are provided as a Source Data file.

cells including monocytes, mature macrophages, microglia and granulocytes[28]. Deletion of ASK1 in macrophages was confirmed by western blot analysis. By contrast, similar ASK1 protein levels were retained in spleen and liver (Supplementary Fig. 3a). We detected slightly reduced ASK1 protein levels in brain, WAT and BAT, which may be explained by the presence of resident myeloid-derived immune cells in these organs, such as microglia in the brain and monocytes and macrophages in BAT and WAT. To test the physiological significance of myeloid cell-specific ASK1 depletion, ASK1$^{F/F}$ and ASK1$^{\Delta mye}$ mice were fed either a chow or a HFD for 20 weeks. As expected, HFD-fed mice revealed significantly higher body weight and impaired glucose tolerance compared to chow-fed mice. Regardless of diets, no difference in body weight and amount of body fat was found between both genotypes (Supplementary Fig. 3b, c). Similarly, myeloid-specific ASK1 knockdown affected neither glucose tolerance nor insulin sensitivity in both diet groups (Supplementary Fig. 3d, e). Taken together, targeted genetic disruption of ASK1 signaling in adipocytes, but not in myeloid cells, elicits beneficial effects on obesity and obesity-induced insulin resistance.

**Increased heat production in HFD-fed ASK1$^{\Delta adipo}$ mice.** Next, we aimed to elaborate on the mechanisms contributing to blunted body weight gain in HFD-fed ASK1$^{\Delta adipo}$ mice (Fig. 2b, c). Importantly, the lean phenotype in ASK1$^{\Delta adipo}$ mice was neither attributable to reduced food consumption nor to increased locomotor activity (Supplementary Fig. 4a, b). In addition, it does not result from impaired intestinal nutrient absorption capacity since fecal caloric content was rather decreased in knockout mice, suggesting an increased energy absorption (Supplementary Fig. 4c). However, we found a significant increase in oxygen consumption rate at room temperature (RT) during both light

and dark phase in adipocyte-specific ASK1 deficient mice (Fig. 3a). Of note, such housing temperature may trigger browning of adipose tissue as it induces chronic thermal stress inducing fuel consumption[29]. Besides browning, elevated heat loss due to fur disruption may affect energy expenditure at RT[30]. Therefore, indirect calorimetry was assessed under thermoneutral conditions, which also revealed similar difference in oxygen consumption between HFD-fed control and knockout mice as observed under RT (Fig. 3b), indicating absence of fur disruption in knockout mice. As the influence of activated BAT on energy expenditure may be negligible under thermoneutral conditions, this finding further suggests that browning of adipose tissue may be crucially involved in the observed phenotype. In line with increased oxygen consumption, heat production (Fig. 3c) as well as rectal temperature (Fig. 3d) were elevated HFD-fed ASK1$^{\Delta adipo}$ mice.

**Increased adipose tissue browning in HFD-fed ASK1$^{\Delta adipo}$ mice.** Analysis of adipose tissue mass at necropsy revealed significantly lower WAT (inguinal, epididymal, mesenteric, and retroperitoneal) as well as BAT fat pad weights in HFD-fed ASK1$^{\Delta adipo}$ mice compared to control littermates (Fig. 4a and Supplementary Fig. 4d) leading to a ~30% reduction of whole-body fat content in ASK1$^{\Delta adipo}$ mice (Fig. 4b). Increased heat production/energy expenditure may suggest activation of BAT or browning of WAT in HFD-fed ASK1$^{\Delta adipo}$ mice. Macroscopically, inguinal WAT appeared darker in ASK1$^{\Delta adipo}$ compared to ASK1$^{F/F}$ mice (Fig. 4c), and immunohistochemical analysis revealed increased positive UCP1 staining in inguinal WAT of knockout mice (Fig. 4d) suggesting increased browning. Moreover, mRNA expression of *Ucp1* and other browning markers in WAT[7] such as *Cidea*, *Pgc1α*, and *Prdm16* were elevated in inguinal fat depots of knockout mice (Fig. 4e). Accordingly,

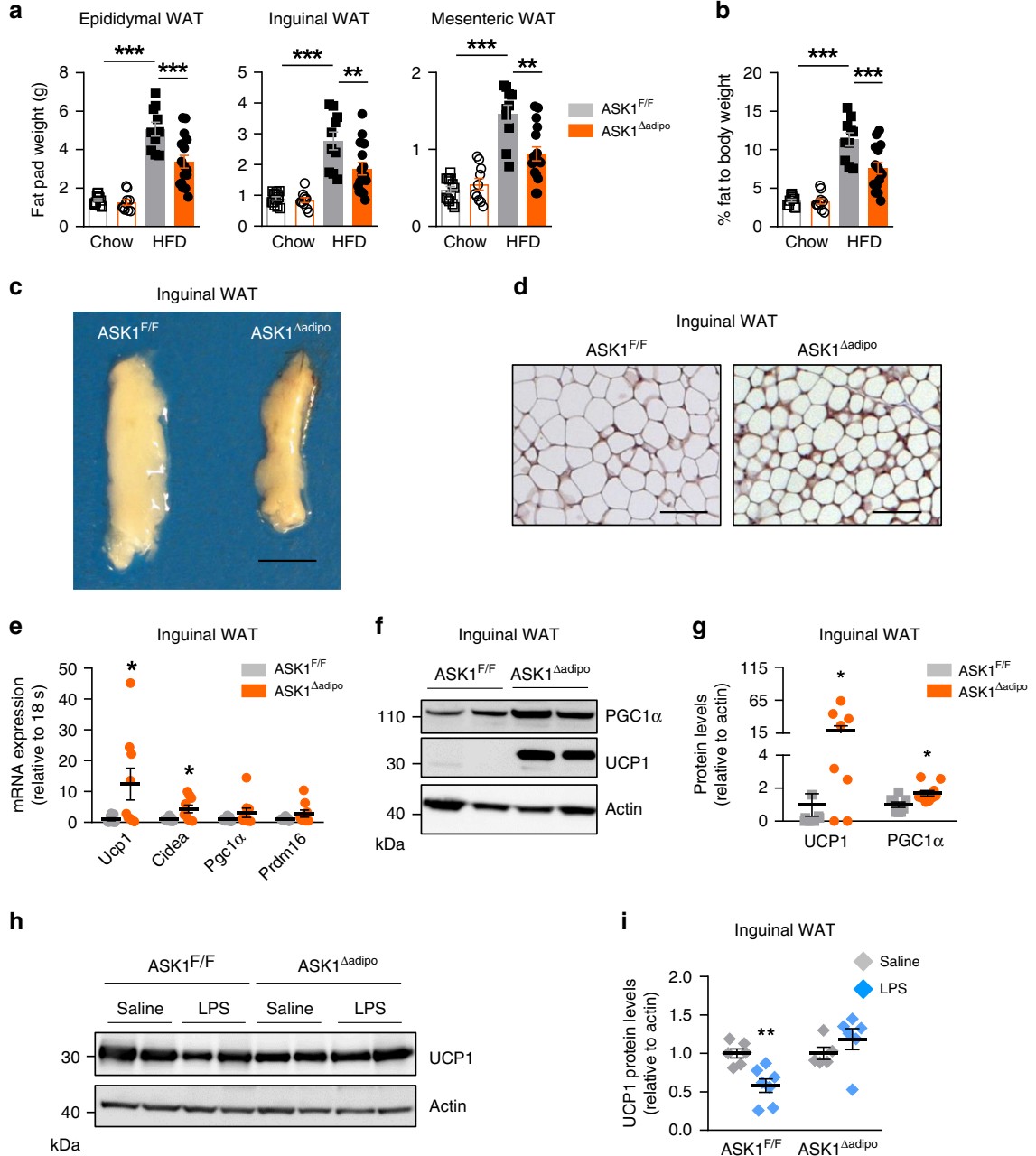

**Fig. 4 Increased adipose tissue browning in HFD-fed ASK1^Δadipo mice. a, b** Fat pad mass as well as total body fat mass (chow-fed: ASK1^F/F, $n = 13$; ASK1^Δadipo, $n = 10$; HFD-fed: ASK1^F/F, $n = 10$; ASK1^Δadipo, $n = 16$) in mice at 18 weeks of age. **$p = 0.005$; ***$p < 0.001$. **c** Representative image of inguinal adipose tissue of a HFD-fed ASK1^F/F and ASK1^Δadipo mouse (scale bar: 10 mm). **d** Representative immunohistochemistry for UCP1 in inguinal adipose tissue of a HFD-fed ASK1^F/F ($n = 3$ mice) and ASK1^Δadipo mouse ($n = 3$ mice) (scale bar: 50 μm). **e** mRNA expression of respective genes in inguinal adipose tissue of HFD-fed ASK1^F/F ($n = 8$) and ASK1^Δadipo ($n = 9$) mice. *$p = 0.027$ (Ucp1), *$p = 0.028$ (Cidea). **f, g** Representative western blot and quantification of UCP1 and PGC1α protein levels in inguinal adipose tissue harvested from HFD-fed ASK1^F/F ($n = 8$) and ASK1^Δadipo ($n = 10$) mice. *$p = 0.031$ (Ucp1), *$p = 0.015$ (Cidea). **h, i** Representative western blot and quantified protein levels of UCP1 in total lysates of inguinal white adipose tissue harvested from cold-exposed mice chronically treated with or without LPS (saline ASK1^F/F, $n = 6$ mice; saline ASK1^Δadipo, $n = 5$ mice; LPS ASK1^F/F, $n = 7$ mice; LPS ASK1^Δadipo, $n = 6$ mice). **$p = 0.003$. Values are expressed as mean ± SEM. Statistical tests used: two-sided $t$-tests for, **e** (Cidea), **i** two-sided Mann–Whitney for **e** (UCP1), **g** ANOVA for (**a** and **b**). Source data are provided as a Source Data file.

western blot analysis of inguinal fat revealed significant higher proteins level of UCP1 and PGC1α in HFD-fed ASK1^Δadipo compared to ASK1^F/F mice (Fig. 4f, g). By contrast, no differences in mRNA expression and protein levels of UCP1 and Cidea were detected in classical BAT (Supplementary Fig. 4e, f). Thus, targeted deletion of ASK1 in adipocytes induces browning of WAT. By contrast, BAT UCP1 protein levels were similar

between the two genotypes indicating that ASK1 depletion may not affect thermogenic potential of classical BAT (Supplementary Fig. 4f).

As shown above (Fig. 1a, b), chronic LPS administration reduced UCP1 protein levels in inguinal adipose tissue of cold-exposed C57BL/6 mice. In order to test a potential involvement of ASK1 in LPS-mediated inhibition/suppression of UCP1, osmotic

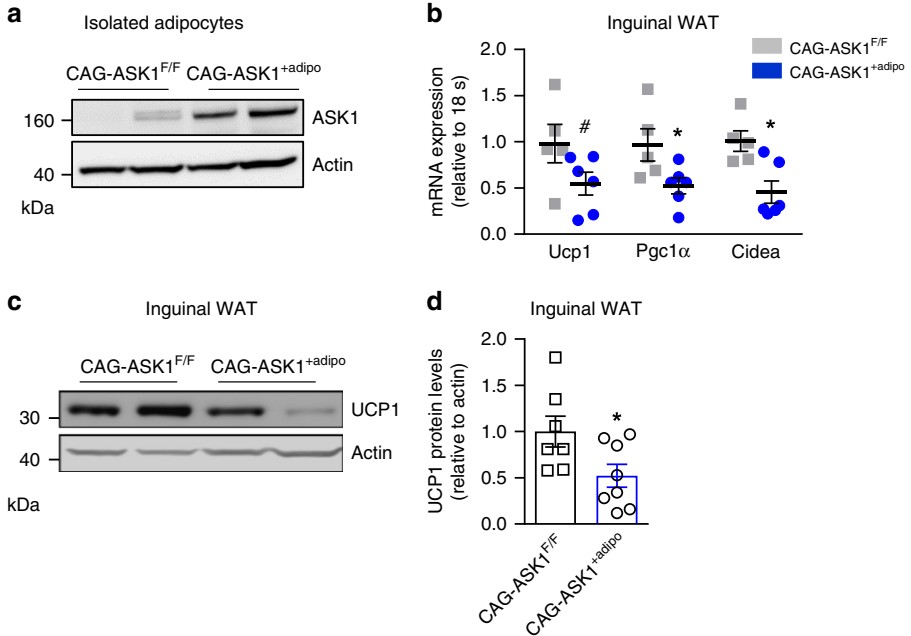

**Fig. 5 Adipocyte-specific ASK1 overexpression decreases adipose tissue browning. a** Protein levels of ASK1 in lysate of isolated adipocytes harvested from CAG-ASK1[F/F] and CAG-ASK1[+adipo] mice ($n = 2$ mice per group). **b** mRNA expression of respective genes in inguinal adipose tissue of cold-exposed CAG-ASK1[F/F] ($n = 5$) and CAG-ASK1[+adipo] ($n = 6$) mice. #$p = 0.094$, *$p = 0.042$, *$p = 0.017$. **c, d** Representative western blot and quantification of UCP1 protein levels in inguinal adipose tissue harvested from cold-exposed CAG-ASK1[F/F] ($n = 7$) and CAG-ASK1[+adipo] ($n = 8$) mice. *$p = 0.037$. Values are expressed as mean ± SEM. Statistical tests used: two-sided $t$-tests for (**b**) (Ucp1 and Pgc1α), **d** two-sided Mann–Whitney for (**b**) (Cidea). Source data are provided as a Source Data file.

mini-pumps releasing either LPS or saline were implanted into ASK1[F/F] and ASK1[Δadipo] littermate mice. Thereafter, mice were cold-exposed for 7 days to trigger browning of WAT[24]. As intended, plasma endotoxin levels in LPS-treated ASK1[F/F] and ASK1[Δadipo] mice were similar between the two groups but significantly elevated compared to saline-treated mice (Supplementary Fig. 4g). While LPS-treatment significantly reduced UCP1 protein levels in cold-exposed ASK1[F/F] mice, UCP1 protein content was maintained in LPS-treated ASK1[Δadipo] mice (Fig. 4h, i). Of note, UCP1 protein levels were not affected in untreated cold-exposed chow-fed ASK1[Δadipo] mice (Supplementary Fig. 4h). Hence, disruption of ASK1 signaling in (inguinal) adipocytes prevents LPS-mediated downregulation of cold-induced UCP1 protein levels.

**Adipocyte-specific ASK1 overexpression decreases adipose tissue browning**. To further investigate a regulatory role of ASK1 in browning of WAT, mice with adipocyte-specific overexpression of ASK1 were generated (CAG-ASK1[+adipo]). After Cre-lox mediated excision of the stop cassette, such mice express ASK1 under the ubiquitous beta actin promoter CAG[31]. Accordingly, adiponectin promoter-driven Cre expression promoted overexpression of ASK1 in white adipocytes (Fig. 5a). Similarly, ASK1 protein levels were increased in BAT whereas ASK1 protein expression was not affected in muscle, brain and liver of CAG-ASK1[+adipo] compared to control littermates (CAG-ASK1[F/F]) (Supplementary Fig. 5a). In parallel to increased ASK1 protein levels, phosphorylated levels of ASK1 (Thr 845) were increased in inguinal adipose tissue of CAG-ASK1[+adipo] mice indicating increased activation of ASK1 (Supplementary Fig. 5b). To induce browning of WAT, chow-fed CAG-ASK1[+adipo] and littermate CAG-ASK1[F/F] mice were cold-exposed for 7 days[24]. Importantly, adipocyte-specific ASK1 overexpression resulted in reduced expression of the browning markers *Ucp1, Cidea* and *Pgc1α* in inguinal adipose tissue (Fig. 5b). In line, protein levels of UCP1

were reduced in inguinal WAT of cold-exposed CAG-ASK1[+adipo] compared to control mice (Fig. 5c, d). By contrast, BAT UCP1 protein levels did not significantly differ between CAG-ASK1[F/F] and CAG-ASK1[+adipo] mice (Supplementary Fig. 5c, d), further supporting the notion that ASK1 signaling affects UCP1 protein levels in WAT but not in BAT.

**ASK1 phosphorylates IRF3 thereby reducing Ucp1 expression**. Next, we aimed to investigate how ASK1 reduces *Ucp1* expression in white adipocytes. Interestingly, ASK1 is required for LPS-mediated phosphorylation and subsequent nuclear translocation of the transcription factor interferon regulatory factor 3 (IRF3)[32]. Moreover, overexpression of IRF3 has been shown to reduce *Ucp1* expression in adipocytes[33], indicating that ASK1-mediated phosphorylation of IRF3 might decrease *Ucp1* expression. Indeed, phosphorylation of IRF3 was significantly reduced in isolated adipocytes of ASK1[Δadipo] mice (Fig. 6a, b). To test whether ASK1 can directly phosphorylate IRF3, we performed an in vitro kinase assay. Indeed, active ASK1 phosphorylated IRF3 in vitro (Fig. 6c, d). Moreover, overexpression of wild-type ASK1 in subcutaneous adipocytes increased IRF3 phosphorylation compared to cells expressing kinase-negative ASK1 (Fig. 6e, f). Next, the role of siRNA-mediated ASK1 knock down on LPS-induced IRF3 phosphorylation was assessed. As intended, ASK1 protein levels were significantly reduced in subcutaneous adipocytes treated with siRNAs targeting ASK1 (Supplementary Fig. 6a). Importantly, LPS-induced IRF3 phosphorylation was significantly blunted in ASK1-depleted adipocytes (Fig. 6g, h) further indicating a contribution of ASK1 to IRF3 phosphorylation. To investigate whether IRF3 is involved in the LPS-mediated downregulation of *Ucp1* expression in adipocytes (Fig. 1d, h), we generated a lentiviral construct expressing a short hairpin RNA (shRNA) against IRF3 (shIRF) to knockdown IRF3 in subcutaneous adipocytes. As expected, expression of IRF3 was significantly reduced in shIRF3 transduced cells when compared

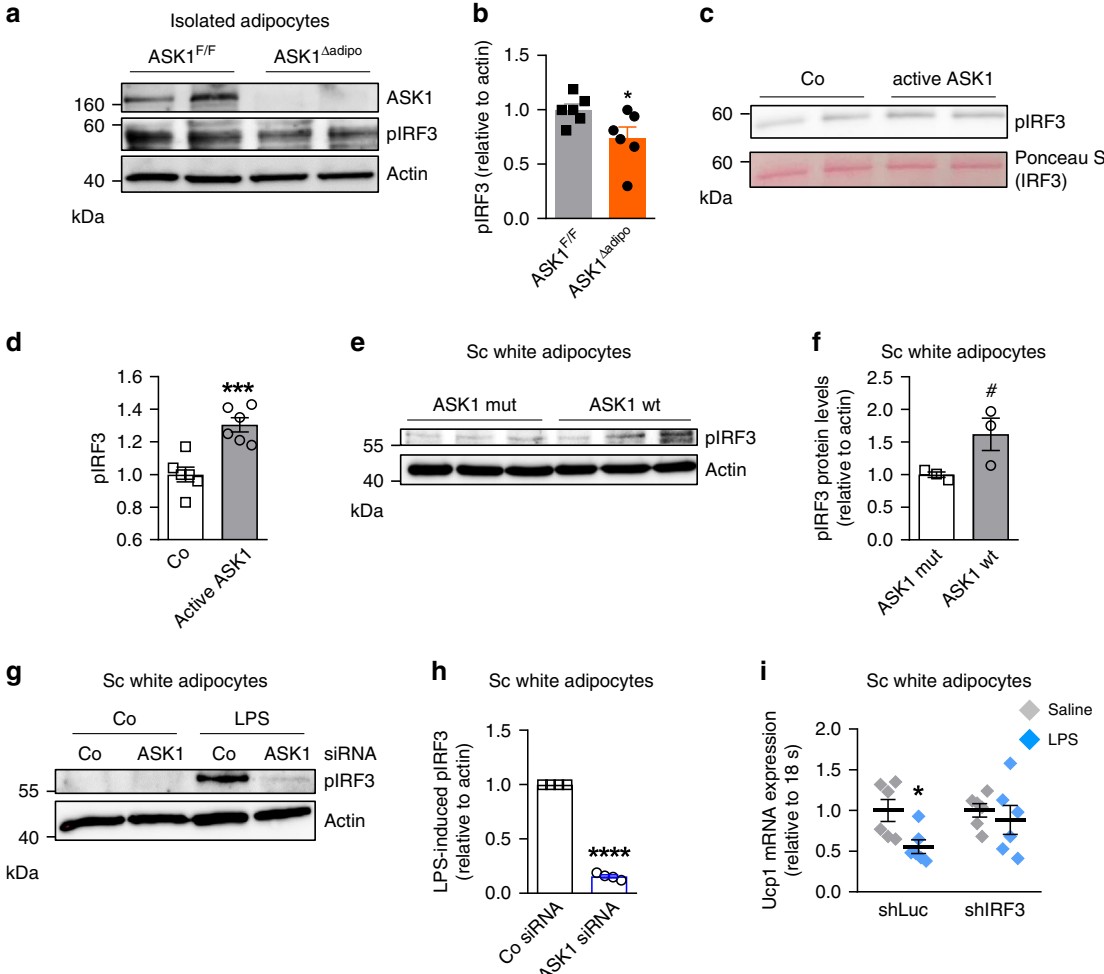

**Fig. 6 ASK1 phosphorylates IRF3. a, b** Representative western blot and quantification of pIRF3 protein levels in isolated adipocytes of HFD-fed ASK1[F/F] and ASK1[Δadipo] mice. $n = 6$ mice per group. *$p = 0.048$. **c, d** Representative western blot and quantification of pIRF3 upon incubation of recombinant IRF3 with or without active ASK1. $n = 6$ biological replicates. ***$p = 0.0006$. **e, f** Western blot and quantification of pIRF3 protein levels in subcutaneous adipocytes transfected with lentivirus expressing wild-type (pLV-CMV-ASK1; ASK1 wt) or kinase-negative (pLV-CMV-ASK1-K716R; ASK1 mut) ASK1. $n = 3$ biological replicates. #$p = 0.070$. **g, h** Representative western blot and quantification of pIRF3 protein levels in subcutaneous adipocytes transfected with control siRNA (Co siRNA) or siRNA targeting ASK1 (ASK1 siRNA) treated with or without 100 ng/ml LPS for 6 h. $n = 4$ biological replicates. ****$p < 0.0001$. **i** Ucp1 mRNA expression in subcutaneous adipocytes transfected with control shRNA lentivirus (shLuc) or shRNA lentivirus targeting IRF3 (shIRF3) pre-treated with 100 ng/ml LPS for 24 h followed by stimulation with 0.1 μM isoproterenol for 6 h. $n = 6$ biological replicates. *$p = 0.019$. Values are expressed as mean ± SEM. Statistical tests used: two-sided $t$-tests for (**b, d, f, i**), two-sided one sample $t$-test for (**h**). Source data are provided as a Source Data file.

to shLuc transfected control cells (Supplementary Fig. 6b). Importantly, IRF3 knockdown significantly blunted LPS-mediated downregulation of Ucp1 expression (Fig. 6i). Similar results were obtained in subcutaneous adipocytes after transfection with siRNAs targeting Irf3 (Supplementary Fig. 6c, d). Such data further confirm the notion that IRF3 is involved in the control of Ucp1 expression in adipocytes.

## Discussion
Browning of WAT significantly contributes to whole-body energy expenditure[34]. Accordingly, identification of molecular pathways that modulate browning has gained high interest since they may serve as pharmacological target to counteract obesity and associated metabolic disorders. Herein, we provide evidence that adipocyte-expressed ASK1 is a negative modulator of WAT browning in obesity thereby affecting energy expenditure, body mass and glucose metabolism. In fact, ASK1 levels were elevated in subcutaneous (inguinal) adipose tissue of HFD compared to

chow-fed mice, and HFD-fed mice with adipocyte-specific ASK1 depletion depicted increased browning of inguinal WAT as well as increased energy expenditure, reduced body weight, blunted liver steatosis and improved glucose metabolism. In support of a regulatory role of ASK1 in adipose tissue browning, adipocyte-specific ASK1 overexpression blunted cold-induced browning of inguinal adipose tissue in chow-fed mice.

Physiologically, increased energy consumption stimulates thermogenesis in brown and/or beige adipocytes via activation of the sympathetic nervous system and/or via secretion of humoral factors[35,36]. Accordingly, high calorie intake may trigger a compensatory adaptation leading to increased energy expenditure, thereby preventing diet-induced obesity. However, diet-induced thermogenesis is unchanged or even blunted in obesity[37] indicating that such compensation may be disturbed under obese condition. Accordingly, identification of targets that block diet-induced thermogenesis in obesity is of great importance. In this regard, PPARγ expression in neurons was identified as a "brake" of high fat diet-induced energy expenditure. In fact, PPARγ

deletion in vagal neurons promoted HFD-induced thermogenesis and reduced body weight gain in mice[38]. Similarly, our results suggest that increased ASK1 expression in adipocytes during obesity may act as a "brake" inhibiting diet-induced browning of WAT. What are potential factors inducing ASK1 in adipocytes? Importantly, ASK1 in adipocytes is induced in response to different stimuli and stressors such as TNFα, Fas ligand or the enzyme glucose oxidase, which induces oxidative stress[26]. Moreover, we present herein evidence that the gut microbiota-derived endotoxin LPS induces ASK1 in adipocytes. In obesity, aforementioned stressors and stimuli are all elevated in adipose tissue[13–15,39,40] promoting ASK1 expression in adipocytes. Of note, activation of ASK1 may be needed to mediate its negative effect on browning. In fact, UCP1 levels were not altered in ASK1-depleted chow-fed mice, whereas they were affected in HFD-fed or LPS-administered ASK1-depleted mice. Importantly, both interventions (HFD-feeding, LPS administration) increased ASK1 phosphorylation/activation. In line, reduced UCP1 protein levels in cold-exposed CAG-ASK1$^{+\text{adipo}}$ mice were paralleled by increased levels of phosphorylated ASK1.

Our finding of increased ASK1 protein levels in inguinal WAT of HFD mice is in line with a previous report revealing increased expression of ASK1 in adipose tissue of obese subjects[23]. The increased ASK1 expression was suggested to contribute to elevated adipose tissue inflammation and, consequently, to obesity-induced metabolic disorders[23,26]. While we only observed a slight reduction in adipose tissue inflammation in HFD-fed ASK1$^{\Delta\text{adipo}}$ mice, we cannot rule out that such difference contributed to the observed phenotype, i.e., to reduced hepatic insulin resistance and lipid accumulation. However, reduced liver lipid accumulation in knockout mice more likely resulted from increased consumption of fatty acids to sustain elevated thermogenesis induced by WAT browning and, thus, decreased ectopic lipid deposition[41]. In turn, reduced hepatic steatosis may lead to blunted liver insulin resistance[42]. ASK1 expression in adipose tissue of obese human subjects was not only increased in adipocytes, but also in the stromal vascular fraction, which comprises of macrophages among other cells[23]. However, we found no effect of myeloid-specific ASK1 depletion on body weight gain and glucose metabolism, suggesting that increased ASK1 expression rather in adipocytes than in macrophages impairs glucose and energy homeostasis in obesity.

Intriguingly, gain and loss of function of ASK1 in adipocytes affected UCP1 protein levels in WAT but not in BAT indicating that ASK1 regulates adipose tissue browning without affecting the thermogenic potential of classical BAT. In further support of such notion, glucose uptake during clamps studies was affected in inguinal WAT but no in BAT of HFD-fed ASK1$^{\Delta\text{adipo}}$ mice. Such finding is in line with other studies reporting specific effects of genetic manipulation (overexpression of Prdm16, deletion of Notch, deletion of the MAPK kinase 6 (MKK6), deletion of Laminin α4 (LAMA4)) on WAT browning without affecting BAT function[43–46]. Similarly, HFD-fed IRF3 deficient mice revealed elevated WAT browning compared to control mice, whereas expression of thermogenic genes in BAT was not affected[33]. In vitro, IRF3 overexpression decreased thermogenic markers in white adipocytes, indicating that IRF3 is a negative modulator of adipose tissue browning[33]. Herein, we confirm a regulatory role of IRF3 in the control of Ucp1 expression in adipocytes, as Irf3 knockdown blunted LPS-mediated decrease in Ucp1 expression in adipocytes. Moreover, ASK1 can directly phosphorylate IRF3 and phosphorylation of the latter was blunted in ASK1 depleted white adipocytes, indicating that ASK1 regulates Ucp1 expression IRF3-dependently in white adipocytes. In addition, ASK1-mediated IRF3 phosphorylation in vivo may also be partially regulated via an indirect mechanism. Indeed, the ASK1 down-stream mediators stress-activated protein kinase p38 and JNK are involved in IRF3 phosphorylation/activation[47,48].

Reduced oxygen consumption and blunted UCP1 protein levels in BAT 12 h upon administration of the β3-adrenoreceptor agonist CL316,243 was recently reported in adipocyte-specific ASK1-deficient mice[49]. Such data indicate that ASK1 expression specifically in adipocytes is involved in the acute regulation of UCP1 in BAT. By contrast, our findings suggest that ASK1 expression in BAT has no major impact on UCP1 protein levels in a more chronic setting, i.e. after cold stimulation for 7 days. Of note, browning of WAT was not assessed in adipocyte-specific ASK1 knockout mice in afore mentioned study[49].

In conclusion, our study identifies a previously unknown role for ASK1 in energy metabolism. We provided evidence that adipocyte-specific ASK1 depletion promotes browning of adipose tissue in HFD-fed mice thereby reducing obesity and glucose intolerance. Thus, ASK1 may be a pharmacological target to combat obesity and associated morbidities given its selective impact on WAT browning.

## Methods

**Animals**. Targeted embryonic stem (ES) cell clones with exon 14 (whose deletion will lead to a frameshift) of ASK1 flanked by loxP sites and a FRT-flanked selection cassette were bought from the European Conditional Mouse Mutagenesis Program (EUCOMM). Morula aggregation for producing chimeric mice from ES cells (B6 background, TyrC+) and foster mothers (Tgv, TyrC-) was performed with the help of Dr. Pawel Pelczar (Institute of Laboratory Animal Science, University of Zurich). Subsequently, obtained chimeras were crossed with B6-albino mice (TyrC-). Off-springs with successful germ-line transmission (black mice) were screened for the presence of the inserted loxP sites by PCR. Mice heterozygous for floxed ASK1 were subsequently interbred to produce homozygous mutant mice. Subsequently, the FRT flanked selection cassette was removed by breeding the floxed ASK1 mice with Flp0 deleter mice[50].

To obtain adipocyte-specific ASK1 depletion (ASK1$^{\Delta\text{adipo}}$), homozygous ASK1 floxed mice without selection cassette were bred to mice that express the Cre enzyme driven by the adipocyte-specific adiponectin promoter (C57BL/6; FVB-Tg(Adipoq-Cre)1Evdr/J mice)[51]. Littermate mice with floxed ASK1 but absent Cre-recombinase (Cre) expression were used as controls (ASK1$^{F/F}$). To obtain myeloid-specific ASK1 depletion (ASK1$^{\Delta\text{mye}}$), homozygous ASK1 floxed mice were bred to mice that express the Cre enzyme driven by the myeloid-specific Lysozyme M-promoter (C57BL/6;129-Lys<tm1(Cre)Ifo>) mice[52]. Littermate mice with floxed ASK1 but absent Cre-recombinase (Cre) expression were used as controls (ASK1$^{F/F}$). Adipocyte-specific knockout mice were genotyped by PCR with primers amplifying the Cre transgene (generating 250 bp Cre (Cre forward GCACTGATTTCGACCA GGTT; Cre reverse CCCGGCAAAACAGGTAGTTA) and 324 bp control (actin forward TGTTACCAACTGGGACGACA; actin reverse GAC ATGCAAGGAGT GCAAGA) allele products) and ASK1 (generating 191 bp wild-type and 216 bp "floxed" allele products (ASK1flox forward CCTCAGAGGAACTGAGCTGCCA; ASK1flox reverse TCAGACCAAGGGCCAGAGAGA)). Myeloid specific knockout mice were genotyped by PCR with primers amplifying the Cre transgene (generating 350 bp wild-type and 700 bp Cre allele products (Cre CCCAGAAATGCCAG ATTACG; LysM forward CTTGGGCTGCCAGAATTTCTC; LysM reverse TTACAGTCGGCCAGGCTGAC) and ASK1 (generating 191 bp wild-type and 216 bp "floxed" allele products). Adipocyte-specific ASK1-overexpressing mice were generated in collaboration with the company PolyGene (Rümlang, Switzerland). The cDNA of ASK1 was inserted into the Rosa26 locus together with the ubiquitous beta actin promoter (CAG)[31] and a loxP-flanked stop cassette (CAG-ASK1$^{F/F}$). ASK1 overexpression in adipocytes was induced via Cre-lox mediated excision of the stop cassette by breeding CAG-ASK1$^{F/F}$ mice to afore mentioned Adipoq-Cre mice (CAGASK1$^{+\text{adipo}}$). All mice were housed in a specific pathogen-free environment on a 12-h-light-dark cycle (light on from 7 am to 7 pm) and fed ad libitum with regular chow diet (ProvimiKliba, Kaiseraugst, Switzerland) or high fat diet (HFD) (58 kcal% fat w/sucrose Surwit Diet, D12331, Research Diets). All protocols conformed to the Swiss animal protection laws and were approved by the Cantonal Veterinary Office in Zurich, Switzerland.

**Western blotting**. Tissues or cells were lysed in ice-cold lysis buffer containing 150 mM NaCl, 50 mM Tris-HCl (pH 7.5), 1 mM EGTA, 1% NP-40, 0.25% sodium deoxycholate, 1 mM sodium pyrophosphate 1 mM sodium vanadate, 1 mM NaF, 10 mM sodium β-glycerolphosphate, 100 nM okadaic acid, 0.2 mM PMSF and a 1:1000 dilution of protease inhibitor cocktail (Sigma-Aldrich, Saint-Louis, MI, USA). Protein concentration was determined using a BCA assay (Pierce, Rockford, IL, USA). Equal amounts of protein were resolved by LDS-PAGE (4–12% gel; NuPAGE, Invitrogen, Basel, Switzerland) and electro-transferred onto nitrocellulose membranes (0.2 μm, BioRad, Reinach, Switzerland). Equal protein loading on

membranes was checked by Ponceau S staining. Blots were blocked in tris-buffered saline (50 mM Tris-HCl, 150 mM NaCl) containing 0.1% Tween (TBS-T) supplemented with 5% non-fat dry milk. Membranes were then placed in a 50 ml Falcon tube and incubated overnight at 4 °C with gentle rotation with the respective primary antibody solutions. Antibody-antigen complexes were detected by using the ECL system and detected with the Fuji LAS-3000 image reader (Fujifilm, Tokyo, Japan). The following primary antibodies were used: UCP1, PA1-24894 (ThermoFisher Scientific, Waltham, MA, USA; diluted 1:1000); Pgc1α, AB 3242 and Actin MAB1501 (Millipore, Darmstadt, Germany; diluted 1:5000); ASK1, AB45178 (Abcam, Cambridge, UK; diluted 1:1000), pASK1, sc-109911 (Santa Cruz Biotechnology, Dallas, TX, USA; diluted 1:200), pIRF3 29047 (Cell Signaling, Danvers, MA, USA; diluted 1:1000), GAPDH, 10494-1-AP (Proteintech, Manchester, UK; diluted 1:1000).

**Immunohistochemistry.** Sections of inguinal adipose tissue were cut (4 μm), deparaffinized, and rehydrated. Slides were incubated with 3% hydrogen peroxide to block endogenous peroxidases. For antigen retrieval, slides were submerged in 0.01 mol/L sodium citrate (pH 6.0) and heated to 95–100 °C for 20-40 min. Slides were incubated with 10% goat serum followed by incubation with anti-UCP1 primary antibody (PA1-24894 (Thermofisher Scientific, Waltham, MA, USA) diluted 1:500 at 4° overnight. Primary antibody binding was detected with an anti-rabbit horseradish peroxidase (HRP) Detection Reagent (Cell Signaling, Danvers, MA, USA). Labeling was visualized with (DAB) (Cell Signaling). Slides were counterstained with hematoxylin.

**RNA extraction and quantitative reverse transcription-PCR (RT-PCR).** Total RNA was extracted with the RNAeasy Lipid Tissue Mini Kit (Qiagen, Hilden, Germany). RNA concentration was determined spectrophotometrically using NanoDrop® (Thermofisher). RNA (0.25–1 μg) was reverse transcribed with PrimeScript RT reagent kit using random hexamer primers (Takara, Kusatsu, Japan). TaqMan system was used for real-time PCR amplification. Relative gene expression was obtained after normalization to 18s RNA, using the formula $2^{-\Delta\Delta cp}$. The following primers/probes were used: Tnfα Mm00443258_m1, Il-6 Mm00446190_m1, Il-1β Mm0043422/8_m1, Mcp1 Mm00441242_m1, Il-10 Mm00439614_m1, F4/80 Mm00802529_m1, Ask1 Mm0043883_m1, 18 s 4352930, Ucp1 Mm01244861_m1, Pgc1α Mm01208835_m1, Cidea Mm00432554_m1, Prdm16 Mm00712556_m1 (Applied Biosystems, Rotkreuz, Switzerland).

**Determination of circulating plasma levels.** Plasma insulin was measured using the Ultra-Sensitive Mouse Insulin ELISA Kit (Chrystal Chem, Downers Grove, IL, USA). Plasma cytokine levels were determined using MSD technology (Meso Scale Discovery, Gaithersburg, MD, USA). Plasma adiponectin levels were measured with an ELISA kit (Axxora, San Diego, CA, USA). LPS was measured in blood plasma by using the Limulus Amebocyte Lysate (LAL) Kit (Lonza, Visp, Switzerland) following the manufacturer's instructions. Endotoxin levels are expressed as arbitrary units.

**Liver triglyceride measurement.** Liver triglycerides were measured from 50 mg of liver tissue according to the method of Bligh and Dyer[53] and quantified with an enzymatic assay (Roche Diagnostics, Rotkreuz, Switzerland).

**Chronic LPS infusion.** Mice were anesthetized with isoflurane and an osmotic mini-pump (Alzet Model 1004; Alza, Palo Alto, CA) was implanted into the intraperitoneal cavity through a 3-mm incision in the abdominal wall. The mini-pumps were either filled with saline (0.9% NaCl) or LPS from *Escherichia coli* (055: B5, Sigma, St. Louis, MO) to infuse 300 μg/kg*day for 24 days.

**Glucose and insulin tolerance test.** For intraperitoneal glucose tolerance test (ipGTT) mice were fasted overnight and for intraperitoneal insulin tolerance tests (ipITT) for 3 h. Either glucose (2 g/kg body weight) or human recombinant insulin (1.0 U/kg body weight) was injected intraperitoneally by a gavage needle (0.30 mm (30 G) × 8 mm; BD Micro-Fine, Becton Dickinson, France). Blood glucose concentration was measured with a Glucometer (Accu-Check Aviva; Roche Diagnostik, Rotkreuz, Switzerland) collected from the tail vein at 0, 15, 30, 45, 60, 90, and 120 min.

**Clamp studies.** Hyperinsulinemic-euglycemic clamp studies were performed in freely moving mice[54]. Mice were anesthetized with isofluoran and eye ointment was applied to both eyes (Vitamin A, Bausch & Lomb Swiss AG, Switzerland). A catheter (MRE 025, Braintree Scientific, Braintree, MA, USA) was inserted into the right jugular vein and exteriorized at the neck. Five to seven days after the surgery, clamp was performed in mice that have lost less than 10% of their pre-operative weight. Insulin was infused at a constant rate (18 mU/kg*min) and steady state glucose infusion rate was calculated once glucose infusion reached a more or less constant rate for 15–20 min with blood glucose levels at 4–5 mmol/l. The glucose disposal rate was calculated by dividing the rate of [3-3 H] glucose infusion by the plasma [3-3 H] glucose specific activity. Endogenous glucose production during the clamp was calculated by subtracting the glucose infusion rate from the glucose

disposal rate. In order to assess tissue specific glucose uptake, a bolus (10 μCi) of 2-[1-14 C] deoxyglucose was administered via catheter at the end of the steady state period. Blood was sampled 2, 15, 25, and 35 min after bolus delivery. Area under the curve of disappearing plasma 2-[1-14C] deoxyglucose was used together with tissue-concentration of phosphorylated 2-[1-14C] deoxyglucose to calculate glucose uptake. Accumulation of 2-[1-14C] deoxyglucose was determined in an aqueous tissue extract after homogenization. Phosphorylated 2-[1-14C] deoxyglucose was separated by Poly-Prep columns (#731-6212, BioRad).

**Metabolic cage analysis.** After a stepwise adaptation to single caging, mice were placed individually in air-tight cages designed for metabolic phenotyping in an open-circuit indirect calorimetric system (PhenoMaster, TSE Systems, Bad Homburg, Germany). The sampling interval for each cage was 2 min, with repetition every 18 min. A total of 72 data points for food intake, $O_2$ consumption, and $CO_2$ production were recorded over a 24-h period. Locomotor activity data were measured using a 2-dimensional infrared light-beam. Energy expenditure (EE) and oxygen consumption ($VO_2$) were calculated using the manufacturer's software and values were corrected for body mass.

**Thermoneutrality.** Single caged mice were put in a temperature and lightning controlled climate chamber system (PhenoMaster, TSE Systems). The temperature was then increased to 30 °C for 20 days.

**Chronic cold-exposure and thermoneutrality.** Single caged mice were put in a temperature and lightning controlled climate chamber system (PhenoMaster, TSE Systems). The temperature was then gradually lowered to 7 °C. On day 7 of the cold-exposure, mice were sacrificed and organs were harvested.

**Rectal temperature measurement.** A digital thermometer (ama-digit ad 15th) was used in combination with a stainless probe. The probe was inserted 2 cm into the anal ducts of mice. Temperature was measured at 4 pm.

**Bomb calorimetry.** Faecal energy content was measured by bomb calorimetry by the Mouse Metabolic Evaluation Facility in Lausanne, Switzerland. Briefly, faeces was collected, dried to constant weight (0.001 g) at 60 °C, and energy content of dried feces (kJ/g) was analyzed for each individual mouse using a bomb calorimeter (IKA C200, Staufen, Germany).

**Cell culture.** The generation and characterization of immortalized subcutaneous white pre-adipocytes was described previously[25]. All cells were seeded onto collagen-coated plates and grown in Dulbecco's modified Eagle's medium (DMEM) containing 25 mM glucose supplemented with 10% fetal bovine serum (FBS) and 1% penicillin/streptomycin (all from Invitrogen, Basel, Switzerland) (complete medium, CM) prior to differentiation. Two days post-confluent, subcutaneous white pre-adipocytes cells were treated with a mixture of methylisobutylxanthine (500 μM), dexamethasone (1 μM), insulin (1.7 μM) and rosiglitazone (1 μM) in CM to induce differentiation (day 0, D0). Two days later (D2), medium was changed to high glucose culture medium containing insulin (0.5 μM). Subsequently culture medium was replaced every other day until the day of experiment. Differentiated subcutaneous cells were transfected with 100 nM of siRNA SMARTpools, siRNA targeting ASK1 or non-targeting siRNA SMARTpools, control siRNA for 24 h, as previously described for HepG2 cells[55]. LPS treatment was started 24 after ASK1 siRNA transfection. For IRF3 knockdown, mature cells were reverse transfected by Lipofectamine RNAiMAX (ThermoFisher Scientific) with 50 nM of a pool of 4 different siRNA targeting Irf3 (LQ-041095-00-0005, Dharmacon) or ON-TARGETplus Non-targeting Control Pool siRNA (D-001810-10-50, Dharmacon). 48 h post-transfection, cells were treated with LPS for 24 h. 18 h later, cells were treated with 100 nM ISO for 6 h.

**Vector generation.** A total of 3 siRNA sequences predicted to target the ASK1 or IRF3 protein were designed with the siDESIGN center tool from Dharmacon. The knockdown efficiency of each construct was tested in vitro and the best candidate sequence of both targets was converted into the relative shRNA and sub-cloned down-stream the U6 promoter of the lentiviral vector pLL3.7 using HpaI (5′) and XhoI (3′). As a control, we generated shRNA against luciferase (shLuc). pLV-CMV-eGFP, pLV-CMV-ASK1 (wild-type) and pLV-CMV-ASK1-K716R (mutated) vectors were created by Vector Builder Inc (Chicago, IL, USA).

**Lentivirus production.** In order to produce the relative lentiviruses, HEK-293LTV cells (Cell Biolabs, San Diego, CA, USA) were initially plated into a p15 dish and then transfected when ~ 90% confluent with 9 μg of pMD2G (envelop vector), 24 μg of pPAX2 (packaging vector) and 30 μg of the relative trans-vector in the presence of 1 mg/ml of polyethylenamine (PEI). 12 h after transfection, media were refreshed. 24 h later, media were collected and stored at 4 °C while the cells received new media. This procedure was repeated 24 h later and then media for the same lentivirus were pooled. After spinning at 2000 × g for 10 min at 4 °C to precipitate cells debris, 5X PEG-IT Virus Precipitation Solution (SBI System Biosciences,

Mountain View, CA, USA) was added to the media. After vortexing, media were left at 4 °C overnight (at least 12 h). The day after, media were spun at 1500 × g for 30 min at 4 °C. Pellets (lentivirus particles) were suspended with PBS + 25 mM Hepes in 1/1000 of the initial volume of the pooled media. Aliquots were created and stored at −80 °C. Lentivirus titers were biologically established by qPCR.

**Lentivirus transduction**. Lentiviral stocks were prepared as described above. scWAT adipocytes were reverse transduced with shLuc or shASK1 (12.5 MOI), with shLuc or shIRF3 (20 MOI) lentiviral particles in CM supplemented with polybrene (8 μg/ml). After 6 h, medium was replaced to CM supplemented with 0.5 μM insulin and changed every other day until the day of experiment (Day 11). For experiments overexpressing ASK1, mature cells were reverse transduced with 10 MOI of pLV-CMV-eGFP, pLV-CMV-ASK1 or pLV-CMV-ASK1-K716 in the presence of polybrene (8 μg/ml). Cells were harvested 48 h after eGFP expression was detected.

**LPS treatment of mature subcutaneous adipocytes**. Cells were differentiated as described above for 6 to 7 days, or 11 days for virus transduced cells, and then treated with 100 ng/ml LPS from Escherichia (coli 055: B5, Sigma) for 24 h. The next day, cells were treated with 0.1 μM isoproterenol (Sigma) for 6 h. Subsequently cells were immediately frozen at −80 °C until further processing.

**ASK1 kinase assay**. Recombinant IRF3 (1–2 μg; ab205216, Abcam) was incubated in a reaction buffer (40 mM Tris pH 7.5, 20 mM MgCl$_2$, 50 μM DTT, 200 μM ATP) with or without active ASK1 (50 nM; V3881, Promega, Dübendorf, Switzerland) for 60 min at 30 °C. Total reaction volume was 20 μl. Reaction was stopped by the addition of 20 μl 2x Laemmli sample buffer.

**Data analysis**. Data are presented as means ± SEM. Normally distributed data were analyzed by unpaired two-tailed Student's t test, one-way ANOVA with Tukey or Newman-Keuls correction for multiple group comparisons or two-way ANOVA with Bonferroni multiple comparisons. For non-parametric data the Mann–Whitney test was used. All statistical tests were calculated using the GraphPad Prism 8.0.0 (GraphPad Software, San Diego, CA, USA). P values < 0.05 were considered to be statistically significant. Power calculation analysis was not performed. The evaluator was blinded to the identity of a specific sample as far as the nature of the experiment allowed it.

**Reporting summary**. Further information on research design is available in the Nature Research Reporting Summary linked to this article.

## Data availability

The data supporting the findings of this study are available within the article or its Supplementary Information files. All data are available from the corresponding author upon reasonable request. The source data underlying all Figures and Supplementary Figures are provided as a Source Data file.

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

## Acknowledgements

This work was supported by a grant from the Swiss National Science Foundation (#310030-160129 and #310030-179344 to D.K.), a grant from the Children's Research Centre of the University Children's Hospital Zurich (to S.W.) and a grant from the Israel Science Foundation (#ISF 928-14 to A.R.). We would like to greatly acknowledge Prof. Thomas Lutz, Dr. Tito Borner and Dr. Christina Neuner-Boyle (University of Zurich, Zurich, Switzerland) for sharing their metabolic cage facility to perform the cold-exposure experiments and for expert advice as well as Dr. Richard Zuellig (University Hospital Zurich, Zurich, Switzerland) for expert advice on LPS determination.

## Author contributions

F.C.L. designed and performed experiments and analyzed data. S.W. designed and performed experiments and wrote the manuscript. F.I., S.M., T.D.C., M.B., and Y.B. performed experiments. C.W. gave conceptual advice and supervised experiments. A.R. gave conceptual advice. D.K. designed experiments, analyzed data and wrote the manuscript. All authors reviewed and commented on the manuscript.

## Competing interests

The authors declare no competing interests.
