## [Peer Review File · Nature Communications]

Reviewers' comments:

Reviewer #1 (Remarks to the Author):

This manuscript by Konrad and colleagues addresses the possibility that apoptosis signal-regulating kinase 1 (ASK1) has an adverse effect on thermogenesis in beige adipocytes. Using a variety of animal models, they show that chronic LPS treatment blunts UCP1 expression in subcutaneous adipocyte in an ASK1-dependent manner. ASK1-mediated suppression of thermogenesis in mice on HFD results in decreased energy expenditure and glucose metabolism by increase the phosphorylation of IRF3. Taken together, these data demonstrate that ASK1 might be a potential target to combat obesity.

The strength of this paper is in a diversity of animal models (three genetic mouse models) and the fact there is a concordance between these models. However, there are some concerns, especially when they get into mechanism:

Major concerns:

1. In Figs 2D-H, adipocyte ASK1 knockout mice show improved glucose tolerance and insulin sensitivity. These mice are leaner than the controls, however, so the result doesn't tell us much. These assays should be performed before the divergence in body weight to avoid confounding by the effect of adiposity on glucose metabolism.
2. The authors show decreased phosphorylation of IRF3 in ASK1 deficient adipocytes and also that active ASK1 phosphorylates IRF3 in vitro. These studies use an antibody commonly used in the field directed against Ser396, which is a target site for TBK1 and IKKe. The authors then say that IRF3 contains potential consensus sequences for ASK1, however they do not correspond to Ser396. How is S396 phosphorylation being regulated by ASK1 which targets another serine? One could postulate an indirect mechanism, but the authors do an in vitro phosphorylation reaction with purified reagents. These experiments make no sense.
3. The authors show increased thermogenesis in the inguinal adipose tissue of adipocyte-specific ASK1 knockout mice on HFD by showing increased expression of UCP1. This is insufficient. The authors should include more experiments, such as immunostaining of UCP1 and a cold challenge experiment with body temperature as a readout.
4. In Fig. 6, the authors show that adipocyte-specific deletion of the ASK1-upstream regulator Fas recapitulates the phenotype observed in adipocyte ASK1 knockout mice, indicating that Fas might also suppress thermogenesis through an ASK1/IRF3 signaling pathway. They need to close the loop

here by showing that ASK1 or IRF3 overexpression abrogates the phenotype in adipocyte-specific Fas knockout mice or adipocytes. They also need to look at IRF3 expression/phosphorylation in the Fas KO adipocytes.

Minor concerns:

1. In Figs 1A-B, the authors might use “saline” or “vehicle” instead of “NaCl” to indicate the control group. As written it makes it look like a specific salt treatment.

2. Figs 2K-O should move to the supplemental data.

Reviewer #2 (Remarks to the Author):

In this manuscript, the authors conclude that ASK1 mediates the inhibitory effect of caloric surplus or LPS-treatment under cold conditions on adipose tissue through phosphorylating IRF3 and thereby suppressing UCP1 expression. The authors found elevated energy expenditure, reduced obesity and ameliorated glucose tolerance in adipocyte-specific ASK1 KO (ASK1^{Δadipo}) mice, and this appeared to be attributed to increased adipose tissue browning represented by the increased expression of UCP1. Accordingly, ASK1 was required for LPS-induced downregulation of UCP1 under cold or β3-adrenergic receptor-stimulated conditions in WAT or white adipocytes. This manuscript includes novel and interesting findings that facilitate our understanding of the role of ASK1 in metabolic stress-mediated regulation of white adipose tissue. However, more experimental evidence is required to strengthen the authors' argument.

Major points:

1. The authors appear to think that expression levels, rather than activation states, of ASK1 represent overall activity of ASK1 in WAT and adipocytes. However, this should be experimentally addressed by immunoblotting using the antibody that specifically detects activating phosphorylation of ASK1 in Fig. 1e, f, g and Fig. 6a.

2. In Fig. 4, adipocyte-specific overexpression of ASK1 in mice decreased cold-stimulated induction of UCP1. However, UCP1 expression was slightly reduced also in cold-stimulated ASK1^{Δadipo} mice compared with ASK1^{F/F} mice as shown in Fig. 3l (lanes 1 and 2 versus lanes 5 and 6). The authors should perform the same experiments as those shown in Fig. 4b, c, d using ASK1^{Δadipo} mice instead of CAG-ASK1^{+adipo} mice and address this point.

3. In Fig. 5, phosphorylation of IRF3 by ASK1 was overall weak.

3-1. Particularly, the weak phosphorylation in in vitro experiments (Fig. 5c, d) implies the possibility that unknown kinases contaminated with ASK1 contribute to IRF3 phosphorylation. Therefore, the authors should include kinase-negative ASK1 in the experiment in Fig. 5c, d as a negative control. They should also express wild type and kinase-negative ASK1 in subcutaneous adipocytes using lentiviruses and examine IRF3 phosphorylation.

3-2. In Fig. 5a, pIRF3 should be compared with that in chow-fed control mice. Based on the data in Fig. 1e, which demonstrates obvious induction of ASK1 upon HFD, we expect that HFD strongly induces pIRF3 in ASK1^{F/F} mice, but not in ASK1^{Δadipo} mice.

4. The efficiency of IRF3 knockdown was obviously low as shown in Fig. S5b. The authors should use another shRNA targeting a different sequence of IRF3 or examine whether exogenous expression of shRNA-resistant IRF3 recovers the reduction of Ucp1 mRNA expression in shIRF3-expressed cells treated with LPS/isoproterenol in Fig. 5e.

5. Fig. 6 did not show any evidence that ASK1 was involved in the phenotype of HFD-fed Fas^{Δadipo} mice shown here. First of all, the comparison of ASK1 expression between Fas^{F/F} and Fas^{Δadipo} mice was not sufficient in Fig. 6a. More quantitative analysis is required. However, even if ASK1 expression was indeed reduced in HFD-fed Fas^{Δadipo} mice compared with Fas^{F/F} mice, it does show that Fas is required for ASK1 expression in HFD-fed mice but does not necessarily show that ASK1 functions downstream of Fas in the regulation of adipose tissue browning. Therefore, the authors should examine whether exogenous expression of ASK1 suppresses the phenotype of HFD-fed Fas^{Δadipo} mice shown in Fig. 6b-g.

Minor points:

Full membranes should be shown to exclude the possibility that C-terminally truncated ASK1 was expressed in ASK1 KO mice in Fig. 2a and k.

Point by Point Response to the Reviewers' Comments

Responses to Reviewer #1

We thank the reviewer for his/her constructive comments and suggestions. We are happy to learn that he/she found the diversity of animal models (three genetic mouse models) and the concordance between them as a strength.

Major concerns:

1. In Figs 2D-H, adipocyte ASK1 knockout mice show improved glucose tolerance and insulin sensitivity. These mice are leaner than the controls, however, so the result doesn't tell us much. These assays should be performed before the divergence in body weight to avoid confounding by the effect of adiposity on glucose metabolism.

We now performed ipGTT in HFD-fed mice before the divergence in body weight. After 4 days of HFD feeding, the weight between the genotypes was similar (ASK1^{F/F} (n=11): 21.6±0.7 g vs. ASK1^{Δadipo} (n=8): 21.2±0.5 g; p=0.67). However, glucose tolerance was already improved in ASK1^{Δadipo} mice (Supplementary Fig. 2c). Moreover, we assessed glucose and insulin tolerance after 6 weeks of HFD feeding. Weight in these tested cohort was not (yet) significantly different after 6 weeks of HFD albeit weight tended to be lower in ASK1^{Δadipo} mice (ASK1^{F/F} (n=9): 28.0±0.6 g vs. ASK1^{Δadipo} (n=10): 26.7±0.6 g; p=0.15). In this cohort, AUC for ipGTT and ipITT were significantly reduced in ASK1^{Δadipo} mice (see below). Of note, in previously analyzed cohorts with much higher numbers of mice body weight was significantly lower in ASK1^{Δadipo} mice after 6 weeks of HFD as presented in Fig. 2b (ASK1^{F/F} (n=52) vs. ASK1^{Δadipo} (n=58)). Therefore, we suggest to show only findings after 4 days of HFD where we did not observe any difference in body weight at all between the two genotypes. Taken together, these data suggest that depletion of ASK1 in adipocytes positively affects glucose metabolism independent of body mass. Such fact was now added to the revised manuscript (page 7 of *Results* section).

Area under the curve (AUC) of intraperitoneal glucose (left graph) and insulin tolerance test (right graph) performed in adipocyte-specific ASK1 knockout (ASK1^{Δadipo}) and control littermate (ASK1^{F/F}) mice fed a high fat diet for 6 weeks. Values are expressed as mean ± SEM. *p<0.05 (Student's t test).

2. The authors show decreased phosphorylation of IRF3 in ASK1 deficient adipocytes and also that active ASK1 phosphorylates IRF3 *in vitro*. These studies use an antibody commonly used in the field directed against Ser396, which is a target site for TBK1 and IKKe. The authors then say that IRF3 contains potential consensus sequences for ASK1, however they do not correspond to Ser396. How is S396 phosphorylation being regulated by ASK1 which targets another serine? One could postulate an indirect mechanism, but the authors do an *in vitro* phosphorylation reaction with purified reagents. These experiments make no sense.

Consensus sequences reflect a common, but not exclusive, amino acid sequence that can be targeted by a specific kinase, and kinases are known to also phosphorylate targets other than the predicted consensus sequences. The performed *in-vitro* kinase assay was conducted to exactly address this possibility. It suggests that ASK1 can indeed directly phosphorylate IRF3 on Ser396, even if this Ser residues is not within ASK1's consensus sequence. Correspondingly, we now show that overexpressing wild-type ASK1 in subcutaneous adipocytes increased IRF3 phosphorylation on Ser396 compared to cells expressing kinase-negative ASK1 (Fig. 5e and 5f of revised manuscript). In addition, LPS-induced IRF3 phosphorylation on Ser396 was blunted in ASK1-depleted subcutaneous adipocytes (Fig. 5g and 5h of revised manuscript), further indicating that ASK1 contributes to IRF3 phosphorylation. Nevertheless, in the *in vivo* setting, we cannot rule out that ASK1-mediated IRF3 phosphorylation may also be partially regulated via an indirect mechanism. Indeed, the ASK1 down-stream mediators stress-activated protein kinase p38 as well as JNK are involved in IRF3 phosphorylation/activation (Navarro L et al., *J Biol Chem* 1999; Nociari M et al., *J Virol* 2009). We now discuss such fact in the revised manuscript (page 16 of *Discussion* section). Moreover, we removed the predicted consensus site for the phosphorylation of IRF3 (Supplementary Fig. 5a of previously submitted manuscript) from the revised manuscript, as this may indeed be confusing.

3. The authors show increased thermogenesis in the inguinal adipose tissue of adipocyte-specific ASK1 knockout mice on HFD by showing increased expression of UCP1. This is insufficient. The authors should include more experiments, such as immunostaining of UCP1 and a cold challenge experiment with body temperature as a readout.

Besides increased UCP1 mRNA and protein levels, Cidea mRNA and protein levels of PGC1 α were elevated in inguinal WAT of HFD-fed adipocyte-specific ASK1 knockout mice (Figs. 3i-3k). Moreover, O₂ consumption and heat production were elevated in knockout mice (Figs. 3a-3c). As UCP1 protein levels were not affected in brown adipose tissue, such data support increased thermogenesis in WAT of adipocyte-specific ASK1 knockout mice. We also show immunohistochemistry for UCP1 (Fig. 3h) as well as increased rectal body temperature in HFD-fed ASK1 ^{Δ adipo} mice (Fig. 3d). Taken together, molecular (UCP1, Cidea, PGC1 α) as well as physiological readouts (O₂ consumption, heat production, body temperature) clearly support the notion of increased thermogenesis in white adipose tissue of HFD-fed ASK1 ^{Δ adipo} mice.

4. In Fig. 6, the authors show that adipocyte-specific deletion of the ASK1-upstream regulator Fas recapitulates the phenotype observed in adipocyte ASK1 knockout

mice, indicating that Fas might also suppress thermogenesis through an ASK1/IRF3 signaling pathway. They need to close the loop here by showing that ASK1 or IRF3 overexpression abrogates the phenotype in adipocyte-specific Fas knockout mice or adipocytes. They also need to look at IRF3 expression/phosphorylation in the Fas KO adipocytes.

In our opinion, knockdown of an upstream player (herein Fas) in combination with overexpression of a downstream target (herein ASK1 or IRF3) will not further prove causality since we have already shown that overexpression of ASK1 leads to impaired browning (as shown in Fig. 4). Similarly, it is known that IRF3 overexpression reduces *Ucp1* expression in adipocytes (Kumari M et al., *J Clin Invest* 2016).

As suggested by the reviewer, we now analysed IRF3 phosphorylation in inguinal adipocytes isolated from HFD-fed adipocyte-specific Fas knockout mice. In parallel to significantly reduced ASK1 protein levels (Fas^{F/F}: 1.0±0.1 vs. Fas^{Δadipo}: 0.5±0.1; p<0.05), pIRF3 protein levels were reduced by ~30% (see below). However, such decrease was rather heterogeneous and statistically not significant. While these data further confirm that the ASK1/IRF3 axis may be involved in the Fas-induced suppression of thermogenesis, other downstream mediators of Fas may contribute to the observed phenotype. Based on this notion as well as on the fact that a causal role of the ASK1/IRF3 axis in adipocyte-specific Fas knockout mice may not be proven (see above), we decided to remove Figure 6 from the revised manuscript. Since experiments in adipocyte-specific Fas knockout mice only served as a proof of concept, we believe that removal of data gained in adipocyte-specific Fas knockout mice does not weaken the role of the ASK1-IRF3 axis in adipose tissue browning presented in the current manuscript,

pIRF3 protein levels were determined in inguinal adipocytes isolated from HFD-fed adipocyte-specific Fas knockout (Fas^{Δadipo}) and control littermate mice (Fas^{F/F}). Values are expressed as mean ± SEM.

Minor concerns:

1. In Figs 1A-B, the authors might use “saline” or “vehicle” instead of “NaCl” to indicate the control group. As written it makes it look like a specific salt treatment.

As suggested, we now changed “NaCl” to “saline” in Figures 1a and 1b as well as in Figures 3l, 3m, 5i and Supplementary Figure 4g.

2. *Figs 2K-O should move to the supplemental data.*

Figures 2k-2o were now moved to Supplementary Figure 3.

Responses to Reviewer #2

We thank the reviewer for his/her encouraging comments. We are grateful that he/she found our findings novel and interesting.

Major points:

1. The authors appear to think that expression levels, rather than activation states, of ASK1 represent overall activity of ASK1 in WAT and adipocytes. However, this should be experimentally addressed by immunoblotting using the antibody that specifically detects activating phosphorylation of ASK1 in Fig. 1e, f, g and Fig. 6a.

We now provide evidence of increased pASK1 protein levels in inguinal white adipose tissue of HFD compared to chow-fed mice (Supplementary Fig. 1b). Moreover, pASK1 levels were increased in scWAT adipocytes treated with LPS (Supplementary Fig. 1c) as well as in inguinal WAT of mice receiving chronic administration of LPS (Supplementary Fig. 1d). Hence, HFD-feeding or LPS-stimulation not only increase ASK1 expression but also its phosphorylation/activation.

2. In Fig. 4, adipocyte-specific overexpression of ASK1 in mice decreased cold-stimulated induction of UCP1. However, UCP1 expression was slightly reduced also in cold-stimulated $ASK1^{\Delta\text{adipo}}$ mice compared with $ASK1^{F/F}$ mice as shown in Fig. 3l (lanes 1 and 2 versus lanes 5 and 6). The authors should perform the same experiments as those shown in Fig. 4b, c, d using $ASK1^{\Delta\text{adipo}}$ mice instead of CAG- $ASK1^{+\text{adipo}}$ mice and address this point.

As suggested, we chronically cold-exposed chow-fed $ASK1^{F/F}$ and $ASK1^{\Delta\text{adipo}}$ mice and found similar mRNA expression of *Ucp1* and *Pgc1 α* (see below) as well as similar UCP1 protein levels between the genotypes (Supplementary Fig. 4h).

Ucp1 and *Pgc1 α* mRNA expression in inguinal adipose tissue of cold-exposed chow-fed $ASK1^{F/F}$ and $ASK1^{\Delta\text{adipo}}$ mice. Values are expressed as mean \pm SEM.

These data, which are in agreement with Fig. 3l, suggest that ASK1-depletion does not affect browning of white adipose tissue in chow-fed mice. In contrast, ASK1-depletion increased browning in HFD-fed (Figs. 3i-3k) and LPS-administered mice (Figs. 3l and 3m). As HFD-feeding and LPS administration activate ASK1 (see above), such activation may be needed to mediate the negative effect of ASK1 on adipose tissue browning. As mentioned by the reviewer, adipocyte-specific overexpression of ASK1 in mice decreased cold-stimulated induction of UCP1 (Figs. 4b-4d). Importantly, these mice not only revealed increased ASK1 protein levels (Fig.

4a) but also elevated phosphorylation of ASK1 in inguinal white adipose tissue (Supplementary Fig. 5b) supporting the notion that increased activation of ASK1 may be needed for the negative effects of the latter on adipose tissue browning. We now added such fact to the revised manuscript (page 14 of *Discussion* section).

3. In Fig. 5, phosphorylation of IRF3 by ASK1 was overall weak.

3-1. Particularly, the weak phosphorylation in *in vitro* experiments (Fig. 5c, d) implies the possibility that unknown kinases contaminated with ASK1 contribute to IRF3 phosphorylation. Therefore, the authors should include kinase-negative ASK1 in the experiment in Fig. 5c, d as a negative control. They should also express wild type and kinase-negative ASK1 in subcutaneous adipocytes using lentiviruses and examine IRF3 phosphorylation.

Unfortunately, kinase-negative ASK1 was not available for the used kinase assay (V3881, Promega, Dübendorf, Switzerland). However, purity of the used active ASK1 was >95% as determined by densitometry. Therefore, it seems rather unlikely that contamination with an unknown kinase able to phosphorylate IRF3 was responsible for the observed effect. Rather, our experiments indicate that active ASK1 increases IRF3 phosphorylation *in vitro*. As suggested, we now expressed wild type and kinase-negative ASK1 in subcutaneous adipocytes using lentiviruses and analyzed IRF3 phosphorylation. Importantly, phosphorylation of IRF3 was increased in subcutaneous adipocytes expressing wild-type ASK1 compared to cells expressing kinase-negative ASK1 (Fig. 5e and 5f of revised manuscript). These data support the notion that IRF3 is a downstream target of ASK1. In agreement, LPS-induced IRF3 phosphorylation was blunted in ASK1-depleted subcutaneous adipocytes (Figs. 5g and 5h of the revised manuscript), further indicating that ASK1 contributes to IRF3 phosphorylation. Taken together, performed experiments *in vitro* (kinase assay, subcutaneous adipocytes) and observed reduction in phospho-IRF3 protein levels in adipocytes harvested from adipocytes-specific ASK1 knockout mice indicate an important role of ASK1 in IRF3 phosphorylation.

3-2. In Fig. 5a, pIRF3 should be compared with that in chow-fed control mice. Base on the data in Fig. 1e, which demonstrates obvious induction of ASK1 upon HFD, we expect that HFD strongly induces pIRF3 in ASK1^{F/F} mice, but not in ASK1^{Δadipo} mice.

We now analyzed pIRF3 protein levels in isolated inguinal adipocytes of chow and HFD-fed control mice. As expected, pASK1 protein levels were increased in adipocytes harvested from HFD-fed mice, indicating increased ASK1 activity in the latter. In parallel, pIRF3 protein levels were elevated (see below). Importantly, increased pASK1 protein levels positively correlated with elevated pIRF3 protein levels, further suggesting that ASK1 is an upstream kinase of IRF3.

pASK1 and pIRF3 protein levels were determined in inguinal adipocytes isolated from chow or HFD-fed control mice.

4. The efficiency of IRF3 knockdown was obviously low as shown in Fig. S5b. The authors should use another shRNA targeting a different sequence of IRF3 or examine whether exogenous expression of shRNA-resistant IRF3 recovers the reduction of Ucp1 mRNA expression in shIRF3-expressed cells treated with LPS/isoproterenol in Fig. 5e.

We now knocked down IRF3 using other siRNA sequences leading to a 50% reduction of IRF3. In contrast to the previous experiment where one siRNA sequence targeting Irf3 was expressed from lentiviral vectors, we now reverse transfected cells with a pool of 4 different siRNA targeting Irf3. Similar to the previous experiment, LPS-induced downregulation of Ucp1 expression was significantly blunted after IRF3 knockdown. These data, which further support the notion that IRF3 negatively affects Ucp1 expression in adipocytes, were now added to the revised manuscript (Supplementary Fig. 6c, d).

5. Fig. 6 did not show any evidence that ASK1 was involved in the phenotype of HFD-fed $Fas^{\Delta adipo}$ mice shown here. First of all, the comparison of ASK1 expression between $Fas^{F/F}$ and $Fas^{\Delta adipo}$ mice was not sufficient in Fig. 6a. More quantitative analysis is required. However, even if ASK1 expression was indeed reduced in HFD-fed $Fas^{\Delta adipo}$ mice compared with $Fas^{F/F}$ mice, it does show that Fas is required for ASK1 expression in HFD-fed mice but does not necessarily show that ASK1 functions downstream of Fas in the regulation of adipose tissue browning. Therefore, the authors should examine whether exogenous expression of ASK1 suppresses the phenotype of HFD-fed $Fas^{\Delta adipo}$ mice shown in Fig. 6b-g.

We now quantified ASK1 protein levels in inguinal adipose tissue of $Fas^{\Delta adipo}$ mice and found significantly lower levels (see below). These data are in agreement with our previous finding showing that Fas activation induces ASK1 expression in adipocytes (Haim Y et al., *Mol Metab* 2017).

ASK1 protein levels were determined in inguinal adipose tissue harvested from HFD-fed adipocyte-specific Fas knockout and control littermate mice. Values are expressed as mean \pm SEM. * $p < 0.05$ (Student's t test).

We agree that reduced ASK1 protein levels in HFD-fed $Fas^{\Delta adipo}$ mice do not necessarily mean that ASK1 functions downstream of Fas in the regulation of

adipose tissue browning. However, in our opinion knockdown of an upstream player (herein Fas) in combination with overexpression of a downstream target (herein ASK1) will not further prove causality since we have already shown that overexpression of ASK1 impairs browning (as shown in Fig. 4). Moreover, the generation of such mice and the execution of experiments would take at least 18 months. Based on the notion that a causal role of ASK1 in adipocyte-specific Fas knockout mice may not be proven, we decided to remove Fig. 6 from the revised manuscript. Since experiments in adipocyte-specific Fas knockout mice only served as a proof of concept, we believe that removal of data gained in adipocyte-specific Fas knockout mice does not weaken the role of ASK1 in adipose tissue browning presented in the current manuscript, as the latter only served as a proof of concept.

Minor points:

Full membranes should be shown to exclude the possibility that C-terminally truncated ASK1 was expressed in ASK1 KO mice in Fig. 2a and k.

We now provide full membranes for ASK1 protein detection presented in Fig. 2a and new Supplementary Fig. 3a (former Fig. 2k) (see below).

REVIEWERS' COMMENTS:

Reviewer #1 (Remarks to the Author):

The authors have done a nice job addressing my concerns.

I have no further issues.

Reviewer #2 (Remarks to the Author):

The authors appropriately addressed this reviewer's concerns.